# Improvement of Depressed Mood with Green Tea Intake

**DOI:** 10.3390/nu14142949

**Published:** 2022-07-19

**Authors:** Keiko Unno, Daisuke Furushima, Yuya Tanaka, Takeichiro Tominaga, Hirotomo Nakamura, Hiroshi Yamada, Kyoko Taguchi, Toshinao Goda, Yoriyuki Nakamura

**Affiliations:** 1Tea Science Center, University of Shizuoka, 52-1 Yada, Suruga-ku, Shizuoka 422-8526, Japan; gp1719@u-shizuoka-ken.ac.jp (K.T.); yori.naka222@u-shizuoka-ken.ac.jp (Y.N.); 2Department of Drug Evaluation & Informatics Graduate School of Pharmaceutical Science, University of Shizuoka, Shizuoka 422-8526, Japan; dfuru@u-shizuoka-ken.ac.jp (D.F.); m18153@u-shizuoka-ken.ac.jp (Y.T.); s19807@u-shizuoka-ken.ac.jp (T.T.); s22805@u-shizuoka-ken.ac.jp (H.N.); hyamada@u-shizuoka-ken.ac.jp (H.Y.); 3Faculty of Medicine School of Health Science, Kagoshima University, Kagoshima 890-8544, Japan; 4Faculty of Food and Nutritional Sciences, University of Shizuoka, Shizuoka 422-8526, Japan; gouda@u-shizuoka-ken.ac.jp

**Keywords:** arginine, caffeine, catechin, hypothalamus-pituitary-adrenal axis, inflammation, lipopolysaccharide, neuronal PAS domain protein 4 (*Npas4*), State-Trait Anxiety Inventory, self-rated depression scale, theanine

## Abstract

Being in a prolonged depressed state increases the risk of developing depression. To investigate whether green tea intake is effective in improving depression-like moods, we used an experimental animal model of depression with lipopolysaccharide (LPS) and clarified the effects of green tea on the biological stress response and inflammation in the brain. Regarding the stress reduction effect of green tea, we found that the sum of caffeine (C) and epigallocatechin gallate (E) relative to the sum of theanine (T) and arginine (A), the major components of green tea, or the CE/TA ratio, is important. The results showed that depression-like behavior, adrenal hypertrophy as a typical stress response, and brain inflammation were suppressed in mice fed green tea components with CE/TA ratios of 2 to 8. In addition, the expression of *Npas4*, which is reduced in anxiety and depression, was maintained at the same level as controls in mice that consumed green tea with a CE/TA ratio of 4. In clinical human trials, the consumption of green tea with CE/TA ratios of 3.9 and 4.7 reduced susceptibility to subjective depression. These results suggest that the daily consumption of green tea with a CE/TA ratio of 4–5 is beneficial to improving depressed mood.

## 1. Introduction

Depression is the most common mental illness. Even among those who do not suffer from depression, many people experience depressed or melancholy moods on a daily basis. Being in a prolonged depressed state increases the risk of developing depression. Rather than starting treatment with antidepressants after depression has set in, attempts should be made to improve the depressed state in daily life as a form of preventive medicine, which could contribute to maintaining and improving mental health for many people. It has been suggested that regular daily tea consumption contributes to risk reduction in healthy individuals [1]. It has also been suggested that L-theanine and polyphenols can function through multiple pathways simultaneously to collectively reduce the risk of depression. Furthermore, caffeine has also been reported to be involved in preventing depression [2]. 

Key factors in depression include the excessive activation of the hypothalamus–pituitary–adrenal (HPA) axis, inflammation, a weakened monoaminergic system, reduced neurogenesis/neuroplasticity, and reduced diversity of the microbiome, affecting the gut–brain axis [1]. In the stress response indicated by adrenal hypertrophy due to hyperexcitation of the HPA axis, we have reported that theanine (T) and arginine (A) reduce stress, whereas caffeine (C) and epigallocatechin gallate (EGCG, E) act to counteract their effects [3], which can be assessed by the molar ratio of the sum of C and E to the sum of T and A, or CE/TA ratio. We have also shown that differences in the relative proportions of tea ingredients, according to the CE/TA ratio, affect stress responses in mice and humans [4,5,6]. Therefore, we hypothesized that there is an optimal composition of tea components for improving depression-like moods.

The major components of green tea are EGCG, caffeine, theanine, and arginine, and the composition of these components varies depending on the type of tea (matcha, gyokuro, sencha, bancha, etc.), the quality (high, medium, or low grade), and the brewing conditions, including the temperature of the hot water and the brewing time [7,8,9]. To address this complexity, in this study we evaluated the depression reduction effect of green tea in terms of the difference in composition balance of the major components. For example, high-grade matcha tea is considered to have a CE/TA ratio of 1 to 3 because of its high content of theanine and arginine [4]. Mid-grade sencha, which is high in catechins and caffeine, is considered to have a CE/TA ratio of 4–6 [10]. Low-grade bancha, which contains less theanine, is considered to have a CE/TA ratio of 7–10.

To investigate the effects of green tea on depression-like moods, we used an experimental animal model of depression with lipopolysaccharide (LPS) to observe the effects of green tea on the organism’s stress response and inflammation. EGCG, theanine, and caffeine have each been reported to suppress levels of inflammatory cytokines induced by LPS [11,12,13], but their effects when taken together in green tea have not yet been elucidated. In this study, we conducted an evaluation using LPS-injected mice, which are widely used as a model of depression, and focused on the effects of green tea on HPA axis excitation and inflammation in the brain, to scientifically investigate what kind of green tea may actually prevent progression to depression. We then evaluated the effect of green tea intake on the improvement of depressed mood using a questionnaire in healthy subjects.

## 2. Materials and Methods

### 2.1. Animal Experiment

#### 2.1.1. Animals and Green Tea Component Ingestion

Four-week-old male SAMP10/TaIdrSlc mice, obtained from Japan SLC (Shizuoka, Japan), were bred under conventional conditions in a temperature- and humidity-controlled room with a 12/12 h light/dark schedule (23 ± 1 °C; 55 ± 5% humidity; light period, 8.00–20.00 h). A normal diet (CE-2; Clea Co., Ltd., Tokyo, Japan) and tap water were available ad libitum. All study procedures were reviewed and approved by the University of Shizuoka Laboratory Animal Care Advisory Committee (approval no. 195241, 9 January 2020) and were in accordance with the guidelines of the US National Institutes of Health for the care and use of laboratory animals.

The mice were divided into groups as follows: A control group, an LPS group, and LPS groups that ingested green tea components with different CE/TA ratios. These mice were examined on days 1 and 2 after LPS administration, so 13 groups (8 mice per group), 104 mice in total, were used in the experiment. The mice were allowed to freely ingest green tea ingredients with different CE/TA ratios in drinking water for 6 days. 

#### 2.1.2. Preparation of Depressed Mouse Model and Sucrose Preference Test

Mice were reared in separate cages and given water containing green tea ingredients with different CE/TA ratios for 6 days. A sucrose preference test (SPT) was performed per the reported method [14], with some changes. A bottle of sucrose solution (1% *w*/*v*) was placed along with a bottle of green tea component for 2 days to acclimate the mice to the sucrose solution. These were then replaced with bottles of sucrose solution and water to obtain a baseline value of sucrose preference for 24 h before LPS injection. LPS (O55:B5, L2880, Sigma-Aldrich, Tokyo, Japan) was dissolved in normal saline (0.2 mg/mL). Mice were intraperitonially injected with LPS (0.5 mg/kg). After LPS injection, mice drank water or 1% sucrose ad libitum for 24 h in the 1-day experiment (Figure 1A). The stress response and brain inflammation on the second day of LPS injection were also examined in another set of mice. In the 2-day experiment, mice were left overnight (18 h) with food and water removed and then given water and 1% sucrose for the next 24 h (Figure 1B). The difference in sucrose preference after LPS injection relative to the baseline value was determined by the following equation:Difference (%)=[(sucrose intake after LPS injection/baseline value)−1]×100.

#### 2.1.3. Adrenal Hypertrophy and Thymic Atrophy

It has been previously observed that the adrenal glands are enlarged and the thymus gland is atrophied in stressed mice [15]. The wet weight of the adrenal glands and thymus gland were measured in the mice injected with LPS.

#### 2.1.4. Quantitative Real-Time Reverse Transcription PCR (qRT-PCR)

The mice were anesthetized with isoflurane. The brain was carefully dissected, and the hippocampus was immediately frozen. Real-time PCR was performed on the brain samples to compare the expression changes of each gene. Total RNA was extracted from tissues using a purification kit (NucleoSpin^®^ RNA, 740955; Takara Bio Inc., Shiga, Japan) in accordance with the manufacture’s protocol. The obtained RNA was converted to cDNA using the PrimeScript™ RT Master Mix kit (RR036A; Takara Bio Inc., Shiga, Japan) Real-time quantitative PCR analysis was performed using the PowerUp™ SYBR™ Green Master Mix (A25742; Applied Biosystems Japan Ltd., Tokyo, Japan) and automated sequence detection system (StepOne; Applied Biosystems Japan Ltd., Tokyo, Japan). Relative gene expression was measured by previously validated primers for the tumor necrosis factor-α (TNF-α) [16], interleukin 1 β (IL-1β) [17], and lipocalin-2 (Lcn2) [18], and neuronal PAS domain protein 4 (*Npas4*) genes. The primer sequences are listed in Table 1. cDNA derived from transcripts encoding β-actin was used as the internal control.

### 2.2. Clinical Trials

#### 2.2.1. Participants

In a double-blind, randomized, controlled trial, 81 healthy adult volunteers aged 20 years or older were randomly assigned to powdered green tea group A (41 subjects) or group B (40 subjects) (Table 2). The participants received verbal and written information about the study and signed an informed consent form before entering the study. None of the participants indicated that they had acute or chronic diseases, regularly took medication, or smoked habitually. Participants were instructed to drink mainly the powdered green tea being tested, and to avoid other green tea, coffee, or black tea throughout the experiment. The study was conducted in accordance with the Declaration of Helsinki and Ethical Guidelines for Medical and Health Research Involving Human Subjects (Public Notice of the Ministry of Education, Culture, Sports, Science and Technology and the Ministry of Health, Labor and Welfare, 2021). The study protocol was approved by the Ethics Committee of the University of Shizuoka (No. 3–26). This study was registered in the University Hospital Medical Information Network (UMIN) (registration ID no. UMIN 000046109). The study period was from November to December 2021.

#### 2.2.2. Procedure

Three 1.5 g portions of powdered green tea were taken per day (morning, noon, and evening). The first 5 days of the intervention was a washout period. After the 5-day washout and 2 weeks of green tea consumption, the participants completed self-administered questionnaires. They rated their physical condition on an ordinal scale (5, very good; 4, good; 3, normal; 2, slightly bad; 1, bad). Fatigue was rated on a visual analog scale (VAS; 0 to 10) from no fatigue to very fatigued. Sleep was self-rated on a 5-point scale ranging from very good to poor. Number of awakenings was similarly self-rated.

To assess the physiological stress response, the salivary amylase activity (sAA) level was measured upon waking and before bedtime on Tuesdays, Wednesdays, and Thursdays during the intervention period, using a testing strip with a colorimetric system (Nipro Co., Osaka, Japan) [20]. On Fridays, subjective depressive symptoms were measured using a self-rated depression scale (SDS) [21] and feelings of anxiety using the State-Trait Anxiety Scale (STAI) test (Japanese STAI Form X-1, Sankyobo, Kyoto, Japan). Statistical analyses included between-group comparisons of SDS and STAI and within-group comparisons of changes from baseline values. Salivary amylase activity was averaged for each week for between-group and within-group comparisons.

Energy consumption, resting heart rate, rest time, number of steps, sleep duration, bedtime, and sleep efficiency were measured using a commercial-grade activity monitor (Fitbit Charge 4; Fitbit Inc., San Francisco, CA, USA) on the wrist throughout the day, except during bathing.

### 2.3. Statistical Analysis 

Statistical analysis for the animal experiment was performed using one-way ANOVA, and statistical significance was set at *p* < 0.05. Confidence intervals and significance of differences in means were estimated by using Tukey’s honest significant difference test or Fisher’s least significant difference test. Statistical analysis for the clinical trial was performed using SAS v. 9.4 statistical analysis software (SAS Institute Inc., Cary, NC, USA). A paired *t*-test was performed to compare the differences between pre- and post-intervention, and the Wilcoxon signed-rank test was performed on the data. A *p*-value < 0.050 was considered statistically significant.

## 3. Results

### 3.1. Animal Experiment 

#### 3.1.1. Effect of Tea Components on Sucrose Preference in LPS-Injected Mice

To determine whether the compositional balance of caffeine and EGCG to theanine and arginine (CE/TA ratio) makes a difference in the antidepressant effect of green tea, we investigated the composition of components with a CE/TA ratio of 1, 2, 4, 8, and 12 (Table 3) in LPS-injected mice used as a depression model. The green tea component was dissolved in water and the mice were allowed to ingest it ad libitum for 6 days prior to LPS injection.

In the 1-day experiment, mice were dissected 24 h after LPS injection. The preference for sucrose over water during that period was examined (Figure 1). The antidepressant effect of the green tea component was evaluated by SPT: the amount of sucrose mice drank after LPS injection was compared with that before LPS injection. Since there were individual differences in water intake, the values were expressed relative to the amount of sucrose mice drank before LPS injection. Sucrose intake decreased in the water intake group (CE/TA, 0) compared to before LPS injection, but in the green tea intake groups, sucrose intake increased, suggesting that the green tea component had an antidepressant effect. Significant differences in antidepressant effects were observed in groups consuming green tea with a CE/TA ratio of 4 or higher (Figure 2).

In the 2-day experiment, mice were dissected 2 days after LPS injection (Figure 1). The preference for sucrose over water was examined during the 24 h prior to dissection. On the second day after LPS injection, sucrose intake was not decreased in the groups given water and green tea compared to before LPS injection, suggesting that depressive-like behaviors had returned.

#### 3.1.2. Effect of Tea Components on the Stress Response

In the 1-day experiment, mice injected with LPS had adrenal hypertrophy and an enhanced stress response, except for the mice treated with green tea components with a CE/TA ratio of 2 (Figure 3). In addition, the thymus gland was significantly atrophied in all mice except those treated with green tea components with CE/TA ratios of 1 and 2.

In the 2-day experiment, adrenal hypertrophy was observed in all mice except the LPS-treated mice that received green tea components with CE/TA ratios of 4 and 8. Thymic atrophy was observed in all mice.

#### 3.1.3. Effect of Ingesting Green Tea Components on Hippocampal Inflammation

The expression of pro-inflammatory cytokines such as IL1-β and TNF-α in the hippocampus was markedly increased in mice injected with LPS, whereas the expression tended to be suppressed in mice fed green tea components with CE/TA ratios of 1 and 2 in the 1-day experiment (Figure 4). On the other hand, cytokine expression was increased in mice that ingested green tea components with CE/TA ratios of 4 or higher. In the 2-day experiment, the expression of these cytokines was drastically suppressed in mice fed green tea components with a CE/TA ratio of 4 or higher.

Lcn2, which is mainly secreted from astrocytes and is known to have a role in regulating the development of brain damage [22], was also markedly increased in the hippocampus of mice injected with LPS in the 1-day experiment. Its expression tended to be suppressed in mice fed green tea components with CE/TA ratios of 1 and 2, whereas it was increased in mice fed green tea components with CE/TA ratios of 4 or higher. However, the enhanced expression was remarkably decreased in the 2-day experiment. 

#### 3.1.4. Effect of Green Tea Components on the Expression of Transcription Factor *Npas4* in the Hippocampus

Since decreased expression of *Npas4* has been reported to result in anxiety and depression-like behavior [23], the effects of ingesting green tea components were compared. No change in *Npas4* expression was observed in the 1-day experiment, but a significant decrease was observed in the LPS-injected mice in the 2-day experiment. The decrease was significantly suppressed in mice that ingested green tea components with a CE/TA ratio of 4 (Figure 5).

#### 3.1.5. Summarized Effect of Green Tea Components

Green tea component compositions with CE/TA ratios of 1, 2, 4, 8, and 12 were examined in a mouse model of LPS-induced depression. Although depression-like behavior was not observed on the second day of LPS administration, as judged by the sucrose preference test, the stress responses of the body due to adrenal hypertrophy and thymic atrophy, as well as changes in the expression of genes related to inflammation and stress in the hippocampus, were observed to continue. The stress response and brain inflammation were suppressed by ingestion of green tea components with a CE/TA ratio of 2 or less one day after LPS administration; adrenal hypertrophy and brain inflammation were suppressed by ingestion of components with CE/TA ratios of 4 and 8 two days after LPS administration; and *Npas4* was restored to suppressed expression with ingestion of components with a CE/TA ratio of 4. Taken together, these results suggest that green tea components with CE/TA ratios of 2 to 8 may have antidepressant effects (Table 4).

### 3.2. Clinical Trials

In our human clinical trial, 81 healthy subjects were randomly assigned to two powdered green tea groups, A (CE/TA = 4.7) and B (CE/TA = 3.9), for a 5-day washout period followed by a 2-week intervention (Table 5). The powdered green teas were suspended in hot water for the subjects to drink daily. The powdered green tea group B showed a significant post-intervention decrease in State-Trait Anxiety Inventory (STAI) scores (Table 6). This indicates that the consumption of green tea B reduced the participants’ anxiety. The subjective ratings of the degree of depressive tendencies on the self-rated depression scale (SDS) showed no statistically significant differences between the groups, but both groups showed a significant improvement in depressive tendencies after the intervention. This suggests that consuming powdered green tea (CE/TA = 3.9−4.7) may be effective in preventing depression. Salivary amylase activity upon waking was used as an indicator of stress after the intervention. No statistically significant differences were found between groups or between pre- and post-intervention.

## 4. Discussion

In this study, the effects of green tea on depressed mood were examined based on animal experiments using a depressed mouse model and a clinical study on healthy subjects, to determine whether green tea consumption improves anxiety and depression-like moods. We injected LPS intraperitoneally into mice after feeding them green tea components with different CE/TA ratios for 6 days, and we analyzed changes in depressive behavior by SPT, adrenal hypertrophy, and gene expression of inflammatory cytokines in the hippocampus.

Mice that ingested green tea components with a CE/TA ratio of 2 or less showed inhibited adrenal hypertrophy and brain inflammation on the first day after LPS injection. This result is consistent with previous findings showing that green tea with a CE/TA ratio of 2 or less had a stress-reducing effect in mice that underwent psychosocial stress in the form of confrontational housing conditions [4]. On the other hand, mice that consumed green tea components with a CE/TA ratio of 4 or higher showed increased adrenal hypertrophy and accelerated inflammatory responses in the brain. This was thought to be due to the counteracting effects of caffeine and EGCG against theanine and arginine [3]. Theanine content differs between CE/TA ratios of 1–2 and 4–12 (Table 2), but this difference is negligible, as we confirmed that if the CE/TA ratio is the same, the degree of adrenal hypertrophy is the same, even if there are differences in the amounts of theanine and other components [3].

On the second day of LPS injection, adrenal hypertrophy and brain inflammation were expected to decrease. Interestingly, however, they were suppressed in mice fed green tea components with CE/TA ratios of 4 and 8, unlike the day-1 data. The reason for the change in optimal CE/TA ratio is unknown, but it is possible that the optimal composition changed as the body’s response changed from excitation to inhibition over time after application of the stress load [24]. Therefore, at present, green tea with a CE/TA ratio of 2 to 8 is considered to be effective in suppressing depression. In addition, *Naps4* expression was significantly decreased on the second day after LPS injection, but in mice that ingested green tea components with a CE/TA ratio of 4, no changes in expression were observed even when the mice were injected with LPS. This suggests the potential importance of ingesting green tea components, especially those with a CE/TA ratio of 4. We previously found that theanine intake increases *Npas4* expression, which is decreased during stress [25], suggesting that arginine, caffeine, and EGCG also affect its expression. It has been reported that decreased expression of *Npas4* in the hippocampus is associated with depression and anxiety [23,26], and that rats with higher *Npas4* expression are more stress tolerant [27]. Thus, increased *Npas4* expression after LPS injection may be important in subsequent recovery. From this perspective, it was assumed that green tea with a CE/TA ratio of about 4 may have a mitigating effect on depressive-like moods. Therefore, we investigated the effects of green tea intake in healthy subjects.

After 2 weeks of consuming green tea with a CE/TA of 3.9, the subjects had significantly reduced anxiety compared to before green tea consumption. There were also lower depression questionnaire scores in both green tea groups (CE/TA = 3.9 and 4.7) compared to pre-consumption. Although it is necessary to consider the effects of subjective bias in the case of questionnaires, these results alongside the depressed mouse model indicate that green tea intake can reduce the biological stress response and inflammation in the brain, suggesting that improving depression-like moods with green tea intake is promising.

Although caffeine, theanine, and EGCG have been suggested to suppress the levels of inflammatory cytokines induced by LPS via the Toll-like receptor 4–NF-kB signaling pathway [11,12,13], the effects may differ depending on their coexistence. Theanine also normalizes HPA axis hyperactivity [8], while caffeine and EGCG have been found to antagonize this effect [3]. A CE/TA ratio of 2 or less in green tea is required for excitement immediately after stress [4], but a higher CE/TA ratio is predicted to be required for subsequent depression. Although the accumulation of oxidative injury is also thought to be important in depression [28], consumption of green tea components did not significantly suppress elevated lipid peroxide in the cerebral cortex of LPS-injected mice (data not shown).

Much research has been conducted on the functionality of green tea components [29,30,31,32], and the importance of green tea as a form of preventive medicine has attracted attention. However, differences in the functionality of green tea can occur depending on the type and quality of the tea. Therefore, in this study, we examined the actions of green tea based on the CE/TA ratio, which is the difference between its major components. This study has the following limitations: One is that there may be differences in CE/TA ratios for depressed mood between mice and humans. Another is the limitation of the CE/TA ratio in clinical trials. It is necessary to study green tea with CE/TA ratios other than the CE/TA ratio of 4 to 5 examined in this study. Further studies using green teas with different CE/TA ratios are needed, but the results suggest that consuming green tea with a CE/TA ratio of 4 to 5 improves depressed-like moods; the type of tea corresponding to this is sencha.

## 5. Conclusions

To determine whether green tea consumption improves depressive-like moods and reduces the risk of developing depression, we examined its effects using a mouse model of LPS-induced depression. Green tea was evaluated using the molar ratio of its major components: theanine, arginine, caffeine, and EGCG (CE/TA). The results showed that adrenal hypertrophy associated with the activation of the HPA axis and the increased expression of inflammatory cytokines in the hippocampus were suppressed in mice that consumed green tea with CE/TA ratios of 2 to 8. The expression of *Npas4*, which is downregulated in anxiety and depression, was maintained in mice fed green tea with a CE/TA ratio of 4. These data suggested that there was an optimal CE/TA ratio for improving depressive-like moods, which was around 4. In the clinical study, participants who consumed green tea with CE/TA ratios of 3.9 and 4.7 showed improved depressive mood scores compared to before consumption. These findings indicated that the consumption of green tea with a CE/TA ratio of 4 to 5 can improve depressed mood.

## Figures and Tables

**Figure 1 nutrients-14-02949-f001:**
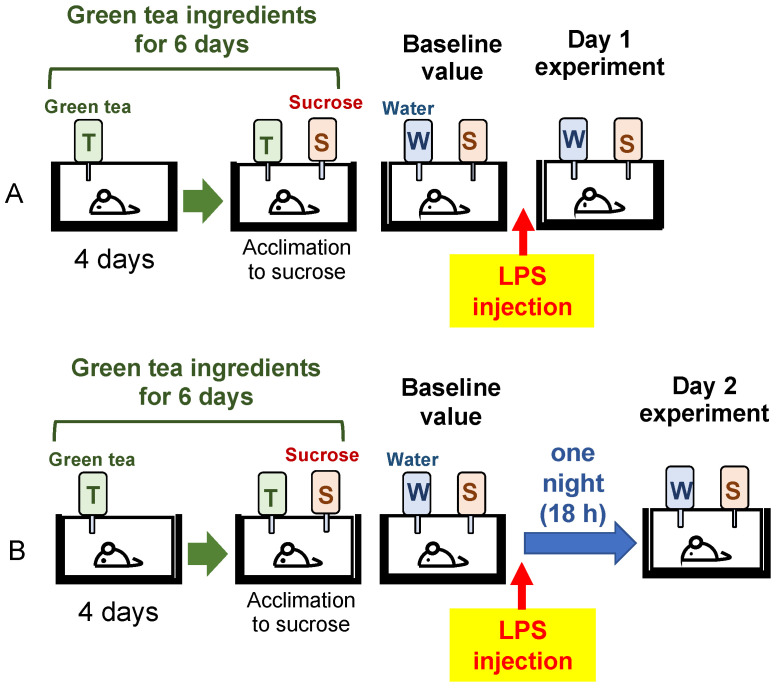
Mice ingested green tea ingredients for 6 days, and a sucrose preference test was performed before and after injection of LPS in (**A**) 1-day and (**B**) 2-day experiment. LPS was dissolved in normal saline (0.2 mg/mL). Mice were intraperitonially injected with LPS (0.5 mg/kg). Bottle labels: T, green tea ingredients; S, 1% (*w*/*v*) sucrose; W, water.

**Figure 2 nutrients-14-02949-f002:**
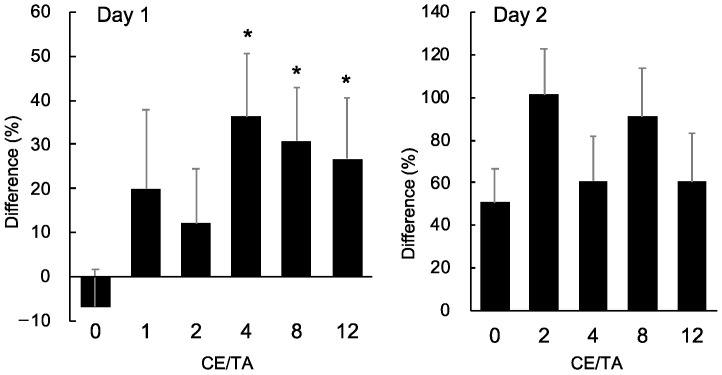
Effect of green tea ingestion on sucrose preference in LPS-injected mice in 1-day and 2-day experiments. Mice drank water or 1% sucrose ad libitum for 24 h. Difference (%) = [(sucrose intake after LPS injection/baseline value) − 1] × 100. Each value represents mean ± SEM (*n* = 8). Asterisks indicate significant differences relative to control (CE/TA = 0) (* *p* < 0.05, Fisher’s least significant difference test).

**Figure 3 nutrients-14-02949-f003:**
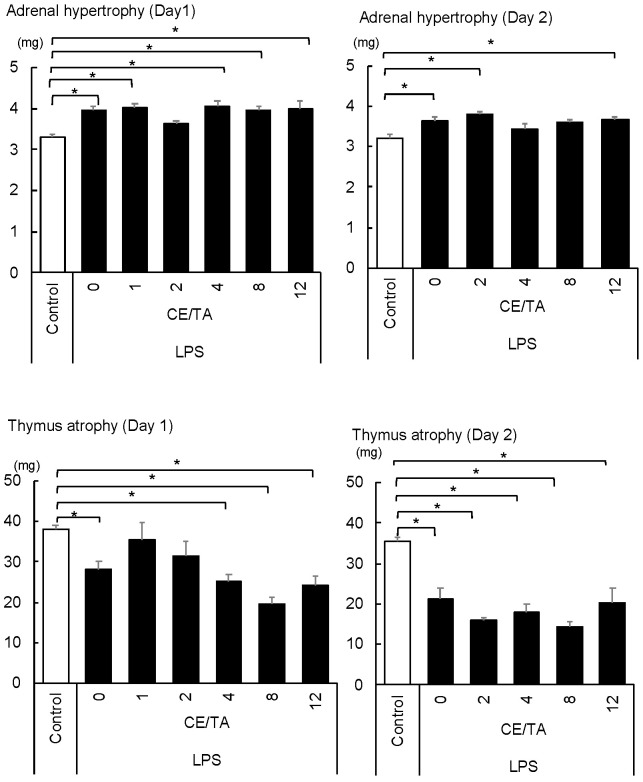
Effect of ingesting green tea component on adrenal hypertrophy and thymic atrophy in LPS-injected mice in 1-day and 2-day experiments. The mice ingested green tea components ad libitum for 6 days prior to LPS injection. Each value represents mean ± SEM (*n* = 8, * *p* < 0.05, Tukey–Kramer method).

**Figure 4 nutrients-14-02949-f004:**
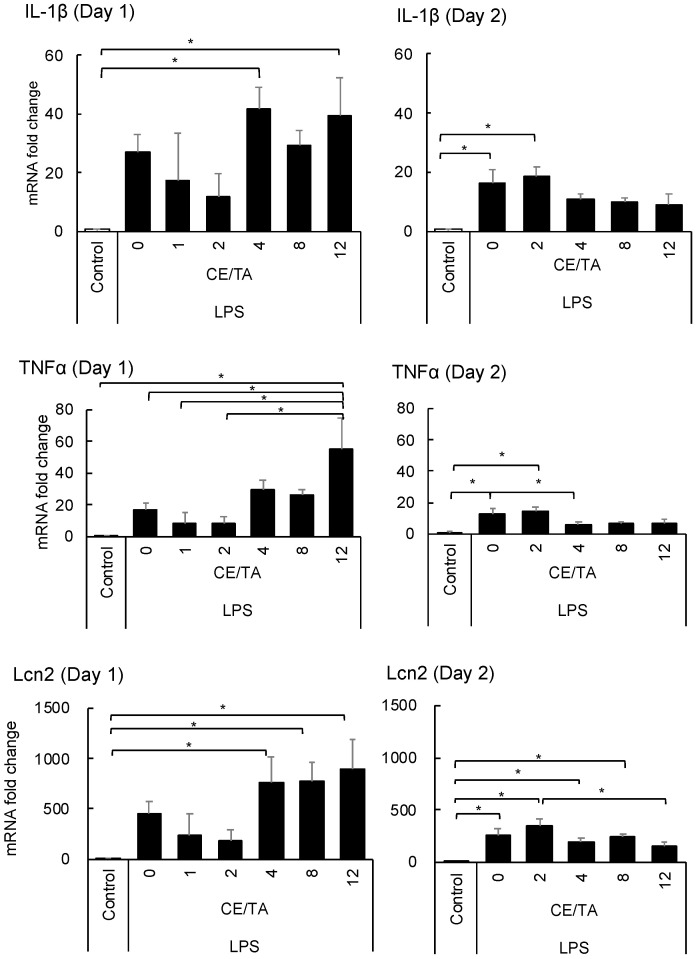
Effect of ingesting green tea components on inflammatory gene expression in the hippocampus of LPS-injected mice in 1-day and 2-day experiments. Mice ingested green tea components ad libitum for 6 days prior to LPS injection. Each value represents the mean ± SEM (*n* = 8, * *p* < 0.05, Fisher’s least significant difference test).

**Figure 5 nutrients-14-02949-f005:**
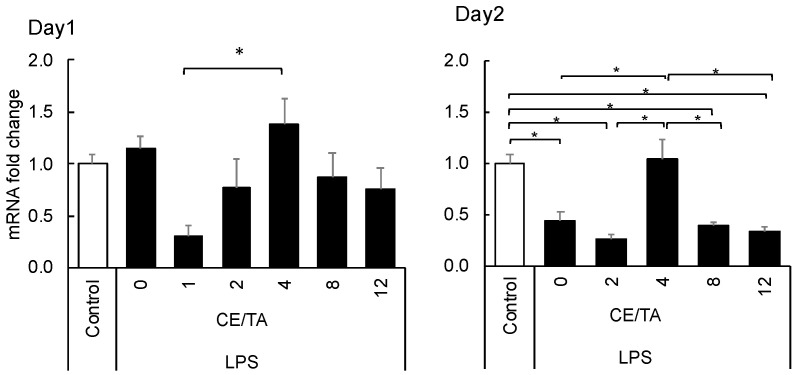
Effect of ingesting green tea components on *Npas4* expression in the hippocampus of LPS-injected mice in 1-day and 2-day experiments. Mice ingested green tea components ad libitum for 6 days prior to LPS injection. Each value represents the mean ± SEM (*n* = 8, * *p* < 0.05, Tukey–Kramer method).

**Table 1 nutrients-14-02949-t001:** Sequence of primers used in qRT-PCR.

Gene	Forward Sequence	Reverse Sequence	Ref.
*TNF-α*	CTGTCTACTGAACTTCGGGGTGAT	GGTCTGGGCCATAGAACTGATG	[16]
*IL-1β*	GCAACTGTTCCTGAACTCAACT	ATCTTTTGGGGTCCGTCAACT	[17]
*Lcn2*	TACAATGTCACCTCCATCCTGG	TGCACATTGTAGCTCTGTACCT	[18]
*Npas4*	AGCATTCCAGGCTCATCTGAA	GGCGAAGTAAGTCTTGGTAGGATT	[19]
*β-actin*	TGACAGGATGCAGAAGGAGA	GCTGGAAGGTGGACAGTGAG	

**Table 2 nutrients-14-02949-t002:** Participants in the clinical trial.

Item	Group A (CE/TA = 4.7)	Group B (CE/TA = 3.9)
Number of participants	41	40
Gender (male/female)	6/35	12/28
Age	52.3 (±15.7)	54.5 (±15.4)
Green tea drinking habit (%)	39 (0.95)	36 (0.90)

**Table 3 nutrients-14-02949-t003:** Mole ratios of theanine, arginine, caffeine, and EGCG.

CE/TA (Mole Ratio)	Theanine (µmole)	Arginine (µmole)	Caffeine (µmole)	EGCG (µmole)
1.0	80	40	60	60
2.0	80	40	120	120
4.0	40	20	120	120
8.0	40	20	240	240
12.0	40	20	360	360

**Table 4 nutrients-14-02949-t004:** Preventive effects of tea consumption on depression-related factors in LPS-treated mice.

Depression-Related Items	Optimum CE/TA	
	1-Day	2-Day
Sucrose preference	4–12	(No depressive behavior)
Adrenal hypertrophy	2	4–8
Thymic atrophy	1–2	No protective effect
Inflammatory gene expression	1–2	4–12
*Npas4* suppression	4	4

**Table 5 nutrients-14-02949-t005:** Powdered green tea components used in the clinical trial.

Powdered Green Tea	Theanine (µmole)	Arginine (µmole)	Caffeine (µmole)	EGCG (µmole)	CE/TA (Mole Ratio)
A	48 ± 1	7 ± 0	120 ± 2	139 ± 2	4.71 ± 0.09
B	53 ± 0	5 ± 0	93 ± 1	136 ± 2	3.94 ± 0.11

**Table 6 nutrients-14-02949-t006:** Effects of green tea ingestion.

Item	Comparison before and after Intervention (*p*-Value)	
	Group A (CE/TA = 4.7)	Group B (CE/TA = 3.9)
State-Trait Anxiety Inventory (STAI)	0.0614	0.0045
Self-rated depression scale (SDS)	0.0410	0.0219
Stress (salivary amylase activity)	0.3307	0.1227
Physical condition	0.0241	0.2262
Subjective sleep sensation	0.6351	0.9070
Number of awakenings	0.7801	0.0631
Energy consumption	0.7382	0.0868
Resting heart rate	0.5200	0.8725
Rest time	0.6305	0.7118
Number of steps	0.6694	0.0963
Sleep time (A)	0.0357	0.0839
Bedtime (B)	0.0734	0.1212
Sleep efficiency (A/B)	0.3503	0.3842

## Data Availability

Not applicable.

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
