# Peer review of "Improvement of Depressed Mood with Green Tea Intake"

_nutrients, 2022, doi:10.3390/nu14142949_

Round 1
Reviewer 1 Report
This paper by Keiko Unno and colleagues looked at the effects of green tea consumption on anxiety and depression-like mood based on animal experiments using a depressed mouse model and a clinical study on healthy subjects. The authors concluded daily consumption of green tea with a CE/TA ratio of 2-5 is beneficial to improving depressed mood. The research question is interesting. But I still have some questions.
Major comments
1. It is very confusing about the mice groups. The authors mentioned eight groups in line 83: “Mice were divided into groups (n = 8): control group, LPS group, and groups of LPS + green tea components with different CE/TA ratios”. However, besides the control group and LPS group, table 2 only showed five groups of “LPS + green tea components with different CE/TA ratios”. Moreover, how many mice were used in total and in each group? Or does each group contain 8 mice? None of these important details were given in the method part.
2. What is the reason for conducting the 1-day experiment and 2-day experiment, and why the groups are different in the two experiments (the 2-day experiment did not contain the CE/TA ratio of 1)?
3. I am not convinced by the summarized effect of green tea components. First, it is very untrustworthy to conclude from Table 3 that green tea components with CE/TA ratios of 1 to 8 have the antidepressant effects, especially since the ratio of 8 was only significant in adrenal hypertrophy in the 2-day experiment among different items. Moreover, based on the animal experiment (‘green tea components with CE/TA ratios of 1 to 8 are expected to have antidepressant effect’) and clinical trial (‘consuming powdered green tea (CE/TA = 3.9–4.7) may be effective in preventing depression’), the authors concluded daily consumption of green tea with a CE/TA ratio of 2-5 is beneficial for improving depressed mood. However, it is not reasonable to combine the suggestive CE/TA ratios from animal experiments and clinical trials.
4. The CE/TA ratio ranges from 1 to 10 in different types and qualities of green tea in the introduction part (lines 59-63). However, the clinical trial only tested participants who consumed green tea with CE/TA ratios of 3.9 and 4.7. Why did the authors only select two groups with similar CE/TA ratios?
Minor comments
5. In lines 54-63, the authors should give the appropriate references.
6. Please give the full name when the abbreviation first appears, such as SPT in line 88.
7. The details of the method, such as the definition of group A and group B, should be explained in the method part instead of the result part.
8. Line 154, the ‘2.2’ should change to ‘2.3’?
9. Line 176, the sentence ‘SPT is an important method to detect anhedonia’ is very abrupt, please delete it.
10. Individual data points should be shown when possible (figures 2-5).
11. Two asterisks in Figure 2 (Day 1: 8 and 12) are very unclear, it seems like the three asterisks were not drawn at the same layer.
12. The authors would benefit from a proofreading service to ensure that the language is correct and precise.
Taken together, I found the concept of this study generally of interest, however, there are many serious issues that prevent this study from being published.
Author Response
Reviewer 1
This paper by Keiko Unno and colleagues looked at the effects of green tea consumption on anxiety and depression-like mood based on animal experiments using a depressed mouse model and a clinical study on healthy subjects. The authors concluded daily consumption of green tea with a CE/TA ratio of 2-5 is beneficial to improving depressed mood. The research question is interesting. But I still have some questions.
Thank you very much for reviewing our manuscript.
Major comments
- It is very confusing about the mice groups. The authors mentioned eight groups in line 83: “Mice were divided into groups (n = 8): control group, LPS group, and groups of LPS + green tea components with different CE/TA ratios”. However, besides the control group and LPS group, table 2 only showed five groups of “LPS + green tea components with different CE/TA ratios”. Moreover, how many mice were used in total and in each group? Or does each group contain 8 mice? None of these important details were given in the method part.
As you indicated, this description was insufficient and has been revised as follows:
Mice were divided into groups as follows: A control group, an LPS group, and LPS groups that ingested green tea components with different CE/TA ratios. These mice were examined on days 1 and 2 after LPS administration, so 13 groups (8 mice per group), 104 mice in total, were used in the experiment. (Line 85-88)
- What is the reason for conducting the 1-day experiment and 2-day experiment, and why the groups are different in the two experiments (the 2-day experiment did not contain the CE/TA ratio of 1)?
We expected that the stress response and brain inflammation would decrease on the second day after LPS administration, but not only that, we found that those responses were different on day 1 and 2 in mice that had consumed green tea components with different CE/TA ratios.
These were added in session 2.1.1 (line 99-101) and in the discussion (line 309-311).
In the day 1 experiment, adrenal hypertrophy and inflammatory responses were significantly different at CE/TA ratio 2 than at CE/TA ratio 4 or higher and control, so data at CE/TA ratio 1 were added for confirmation.
On the other hand, CE/TA ratio 1 was not added to the data in the day 2 experiment because the suppression of adrenal hypertrophy and inflammation was observed at CE/TA ratios of 4 and higher.
- I am not convinced by the summarized effect of green tea components. First, it is very untrustworthy to conclude from Table 3 that green tea components with CE/TA ratios of 1 to 8 have the antidepressant effects, especially since the ratio of 8 was only significant in adrenal hypertrophy in the 2-day experiment among different items. Moreover, based on the animal experiment (‘green tea components with CE/TA ratios of 1 to 8 are expected to have antidepressant effect’) and clinical trial (‘consuming powdered green tea (CE/TA = 3.9–4.7) may be effective in preventing depression’), the authors concluded daily consumption of green tea with a CE/TA ratio of 2-5 is beneficial for improving depressed mood. However, it is not reasonable to combine the suggestive CE/TA ratios from animal experiments and clinical trials.
Thank you for your valuable opinion.
CE/TA ratio 8 showed a significant effect in SPT and an inhibitory trend in inflammatory response(Table 3 and Figure 4 have been revised).
We revised in the discussion and conclusions as follows; These findings suggest that consumption of green tea with a CE/TA ratio of 4 to 5 can improve depressed mood. (Line 350 and line 364-365).
- The CE/TA ratio ranges from 1 to 10 in different types and qualities of green tea in the introduction part (lines 59-63). However, the clinical trial only tested participants who consumed green tea with CE/TA ratios of 3.9 and 4.7. Why did the authors only select two groups with similar CE/TA ratios?
That is certainly true.
We thought we had prepared green teas with CE/TA ratios of 4 and much higher. But in fact, contrary to our expectations, it turned out that the green teas were close in CE/TA ratio, but we did not have time to prepare another green tea, so we compared these two types.
Minor comments
- In lines 54-63, the authors should give the appropriate references.
Some references were added.
- Please give the full name when the abbreviation first appears, such as SPT in line 88.
We added.
- The details of the method, such as the definition of group A and group B, should be explained in the method part instead of the result part.
We revised it. Table 6 in Section 3.2 has been moved to Section 2.2 and made Table 2.
- Line 154, the ‘2.2’ should change to ‘2.3’?
Revised.
- Line 176, the sentence ‘SPT is an important method to detect anhedonia’ is very abrupt, please delete it.
Deleted.
- Individual data points should be shown when possible (figures 2-5).
Sorry, but we could not show individual data in these figures.
- Two asterisks in Figure 2 (Day 1: 8 and 12) are very unclear, it seems like the three asterisks were not drawn at the same layer.
Thank you for your careful checking. Revised them.
- The authors would benefit from a proofreading service to ensure that the language is correct and precise.
Taken together, I found the concept of this study generally of interest, however, there are many serious issues that prevent this study from being published.
We have revised the manuscript and re-edited it according to your valuable suggestions.

Reviewer 2 Report
In this study, Unno et al. analyzed the stress-reducing effects and improvement of depressed mood of green tea in relation to the CE/TA ratio (caffeine C, epigallocatechin gallate E, theanine T and arginine A). In the first part, they analyze an experimental animal model for depression based on LPS injections in mice. Results show suppression of inflammatory cytokines and activation of the HPA axis.
In the second part, a double-blind, randomized controlled trial was conducted and 81 participants consumed green tea with a CE/TA ratio of either 3.9 or 4.7. The results also showed an improvement in depressed mood.
In summary, the studies are well conducted and described, the discussion is well-founded, and the results clearly show an effect on depressed mood.
Only the hypothesis for research is formulated unclearly in the introduction, could be improved and should be refered in the conclusion.
Author Response
Reviewer 2
In this study, Unno et al. analyzed the stress-reducing effects and improvement of depressed mood of green tea in relation to the CE/TA ratio (caffeine C, epigallocatechin gallate E, theanine T and arginine A). In the first part, they analyze an experimental animal model for depression based on LPS injections in mice. Results show suppression of inflammatory cytokines and activation of the HPA axis.
In the second part, a double-blind, randomized controlled trial was conducted and 81 participants consumed green tea with a CE/TA ratio of either 3.9 or 4.7. The results also showed an improvement in depressed mood.
In summary, the studies are well conducted and described, the discussion is well-founded, and the results clearly show an effect on depressed mood.
Only the hypothesis for research is formulated unclearly in the introduction, could be improved and should be referred in the conclusion.
Thank you so much for reviewing our manuscript. The hypothesis was added in the introduction and mentioned it in the conclusion. The introduction was slightly modified accordingly.

Round 2
Reviewer 1 Report
Many thanks for addressing many of my comments, the revised manuscript is much improved.
The authors should include the limitations of this study in the discussion part, such as the limit CE/TA ratios in the clinical trial and the difference of CE/TA ratios on depressed mood between mice and human.
Moreover, the subtitle of the method part is still incorrect. Lines 124 and 146.
Author Response
Many thanks for addressing many of my comments, the revised manuscript is much improved.
Thank you so much for re-reviewing our manuscript.
The authors should include the limitations of this study in the discussion part, such as the limit CE/TA ratios in the clinical trial and the difference of CE/TA ratios on depressed mood between mice and human.
Thank you for your valuable suggestion. We added the limitations in the discussion (line 346-350).
Moreover, the subtitle of the method part is still incorrect. Lines 124 and 146.
We revised.
